# ADALEAD: A SIMPLE AND ROBUST ADAPTIVE GREEDY SEARCH ALGORITHM FOR SEQUENCE DESIGN

## ABSTRACT

Efficient design of biological sequences will have a great impact across many industrial and healthcare domains. However, discovering improved sequences requires solving a difficult optimization problem. Traditionally, this challenge was approached by biologists through a model-free method known as "directed evolution", the iterative process of random mutation and selection. As the ability to build models that capture the sequence-to-function map improves, such models can be used as oracles to screen sequences before running experiments. In recent years, interest in better algorithms that effectively use such oracles to outperform model-free approaches has intensified. These span from approaches based on Bayesian Optimization, to regularized generative models and adaptations of reinforcement learning. In this work, we implement an open-source Fitness Landscape EXploration Sandbox (FLEXS) environment to test and evaluate these algorithms based on their optimality, consistency, and robustness. Using FLEXS, we develop an easy-to-implement, scalable, and robust evolutionary greedy algorithm (AdaLead). Despite its simplicity, we show that AdaLead is a remarkably strong benchmark that out-competes more complex state of the art approaches in a variety of biologically motivated sequence design challenges.

## 1 INTRODUCTION

An important problem across many domains in biology is the challenge of finding DNA, RNA, or protein sequences which perform a function of interest at a desired level. This task is challenging for two reasons: (i) the map $\phi$ between sequences $\boldsymbol{X} = \{\boldsymbol{x}_1, \cdots, \boldsymbol{x}_n\}$ and their biological function $\boldsymbol{y} = \{y_1, \cdots, y_n\}$ is non-convex and (ii) has sparse support in the space of possible sequences $A^L$, which also grows exponentially in the length of the sequence $L$ for alphabet $A$. This map $\phi$ is otherwise known as a "fitness landscape" (de Visser et al., 2018). Currently, the most widely used practical approach in sequence design is "directed evolution" (Arnold, 1998), where populations of biological entities are selected through an assay according to their function $y$, with each iteration becoming more stringent in the selection criteria. However, this model-free approach relies on evolutionary random walks through the sequence space, and most attempted optimization steps (mutations) are discarded due to their negative impact on $y$.

Recent advances in DNA sequencing and synthesis technologies allow large assays which query $y$ for specific sequences $\boldsymbol{x}$ with up to $10^5$ physical samples per batch (Barrera et al., 2016). This development presents an opening for machine learning to contribute in building better surrogate models $\phi' : \boldsymbol{X} \to \boldsymbol{y}$ which approximate the oracle $\phi$ that maps each sequence to its true function. We may use these models $\phi'$ as proxies of $\phi$, in order to generate and screen sequences *in silico* before they are sent to synthesis (Yang et al., 2019; Fox et al., 2007). While a large body of work has focused on building better local approximate models $\phi'$ on already published data (Otwinowski et al., 2018; Alipanahi et al., 2015; Riesselman et al., 2017; Sinai et al., 2017), the more recent work is being done on *optimization* in this setting (Biswas et al., 2018; Angermueller et al., 2020; Brookes & Listgarten, 2018; Brookes et al., 2019). Although synthesizing many sequences within a batch is now possible, because of the labor-intensive nature of the process, only a handful of iterations of learning can be performed. Hence data is often collected in serial batches $b_i$, comprising data $\mathcal{D}_t = \{b_0, \cdots, b_t\}$ and the problem of sequence design is generally cast as that of proposing batches so that we may find the optimal sequence $\boldsymbol{x}_t^* = \arg\max_{\boldsymbol{x} \in \mathcal{D}_t} \phi(\boldsymbol{x})$ over the course of these experiments.

In this paper, we focus our attention on ML-augmented exploration algorithms which use (possibly non-differentiable) surrogate models $\phi'$ to improve the process of sequence design. While the work is under an active learning setting, in which an algorithm may select samples to be labelled, with data arriving in batches $b_i$, our primary objective is black-box optimization, rather than improving the accuracy of surrogate model. We define $\mathcal{E}_\theta(\mathcal{D}, \phi')$ to denote an exploration algorithm with parameters $\theta$, which relies on dataset $\mathcal{D}$ and surrogate model $\phi'$. When the context is clear, we will simply use $\mathcal{E}$ as shorthand.

In most contexts, the sequence space is large enough that even computational evaluation is limited to a very small subset of possible options. For this reason, we consider the optimization as *sample-restricted*, not only in the number of queries to the ground truth oracle, but also the number of queries to the surrogate model(Among other reasons, this allows us to thoroughly study the algorithms on landscapes that can be brute-forced, simulating a similar situation when the sequence space is very large compared to computation power, a very common setting.) The algorithm $\mathcal{E}$ may perform $v$ sequence evaluations in silico for every sequence proposed. For example, $v \times B$ samples may be evaluated by the model before $B$ samples are proposed for measurement. Ideally, $\mathcal{E}$ should propose strong sequences even when $v$ is small; that is, the algorithm should not need to evaluate many sequences to arrive at a strong one.

## 2 CONTRIBUTIONS

In this study, we make three main contributions towards improving algorithms for sequence design:

1. To build on recent progress in biological sequence design, the research community needs good benchmarks and reference algorithm implementations against which to compare new methods. We implement an open-source simulation environment FLEXS that can emulate complex biological landscapes and can be readily used for training and evaluating sequence design algorithms. We hope that FLEXS will help ensure meaningful and reproducible research results and accelerate the process of algorithm development for ML-guided biological sequence design.

2. We introduce an abstracted oracle to allow the empirical study of exploration strategies, independent of the underlying models. This helps us understand relevant properties, such as robustness and consistency of the algorithms.

3. Inspired by evolutionary and Follow the Perturbed Leader approaches in combinatorial optimization, we propose a simple model-guided greedy approach, termed Adapt-with-the-Leader (ADALEAD). ADALEAD is simple to implement and is competitive with previous state-of-the-art algorithms. We propose ADALEAD as a strong, accessible baseline for testing sequence design algorithms. We show that in general, simple evolutionary algorithms are strong benchmarks to compete against and should be included in future analyses of new methods.

## 3 EVALUATION

We evaluate the algorithms on a set of criteria designed to be relevant to both the biological applicability as well as the soundness of the algorithms considered (Purohit et al., 2018). We run the algorithms using FLEXS, where all of these algorithms and criteria evaluators are implemented.

- **Optimization:** We let maximization be the objective. Most optimization algorithms operate under the assumption that critical information such as the best possible $y^*$ or the set of all local maxima $\mathcal{M}$ is unknown. While it is reasonable to assume that the best sequence is the one with the highest fitness, this is not necessarily the case in reality. For instance, we might wish to bind a particular target, but binding it too strongly may be less desirable than binding it at a moderate level. As measurements of this criterion, we consider the maximum $y = \max_{\boldsymbol{x}} \phi(\boldsymbol{x})$ over all sequences considered, as well as the cardinality $|\mathcal{S}|$, where $\mathcal{S} = \{\boldsymbol{x}_i \mid \phi(\boldsymbol{x}_i) > y_\tau\}$ and $y_\tau > 0$ is a minimum threshold value. It is noteworthy that we often do not know if *any* solutions $y > y_\tau$ exists, hence finding many solutions by an algorithm is a sign of its strength.

- **Robustness:** A major challenge for input design in model-guided algorithms is that optimizing directly on the surrogate $\phi'$ can result in proposing a sequence $\boldsymbol{x}$ with large error, instead of approximating $\boldsymbol{x}^*$ (e.g. if the proposed input $\boldsymbol{x}$ is far outside $\mathcal{D}$). Additionally, while biological

models $\phi$ may contain regularities that are universally shared, those regularities are not known. Hence, a desired property of the algorithm is that it is robust in the face of a poor model.

- **Consistency:** The algorithm $\mathcal{E}$ should produce better performing sequences if it has access to a higher quality model $\phi'$.

Additionally, we desire that the high-performing sequences proposed by the algorithm are diverse. Because a sequence may be disqualified for reasons which are unknown during the design phase, we would like to find distinct solutions which meet the optimality criteria. However, metrics for measuring diversity can be ambiguous and we only focus on measuring diversity in a narrow sense. We measure diversity using $|\mathcal{S}|$. When the ground truth model can be fully enumerated to find its local optima (peaks) by brute force (i.e. we know the maxima $\mathcal{M}$), we can use the number of found maxima $|\mathcal{M}'_{y_\tau}|$ as a measure of diversity, where $\mathcal{M}'_{y_\tau} \subset \mathcal{M}$ represents the maxima found by the algorithm above fitness $y_\tau$.

## 4  RELATED WORK

**Bayesian Optimization (BO).** BO algorithms are designed to optimize black-box functions which are expensive to query. Importantly, these algorithms make use of the uncertainty of model estimates to negotiate exploration versus exploitation. In a pioneering study, Romero et al. (2013) demonstrate the use of BO for protein engineering. Many successful attempts have followed since (e.g. Gonzalez et al. (2015) and Yang et al. (2019)), however in each of these cases a specific model of the design space is assumed to shrink the search space significantly. Otherwise BO is known to perform poorly in higher dimensional space (Frazier, 2018), and to our knowledge, no general purpose sequence design algorithm using BO has performed better than the models considered below. For this reason, while we implement variations of BO as benchmarks (see similar interpretations in Belanger et al. (2019)), we do not consider these implementations as competitive standards. In our figures, we use the EI (Expected Improvement) acquisition function with an evolutionary sequence generator as our BO algorithm, and show comparisons with alternatives (on TF landscapes) in the supplement.

**Generative models.** Another class of algorithms approach the task of sequence design by using regularized generative models. At a high level, these approaches pair a generative model $G_\varphi$ with an oracle $\phi'$, and construct a feedback loop that updates $G_\varphi$ (and sometimes $\phi'$) to produce high-performing sequences. In Feedback-GAN (FBGAN), Gupta & Zou (2018) pair a generative adversarial network (Goodfellow et al., 2014) that is trained to predict whether sequences belong to the functional set using a frozen oracle $\phi'$ which filters a subset of sequences at each training step. They bias the training of the generator and discriminator towards high-performing sequences. Killoran et al. (2017) pursue the sequence optimization by regularizing an "inverted model", ${\phi'_\theta}^{-1}(y_i) = x_i$ with a Wasserstein GAN (Arjovsky et al., 2017) which is trained to produce realistic samples. In this case, both $\phi'_\theta$ and the generator are trained jointly.

Brookes & Listgarten (2018) propose an algorithm, Design by Adaptive Sampling (DbAS), that works by training a generative model $G_\varphi$ on a set of sequences $\boldsymbol{X}_0$, and generating a set of proposal sequences $\hat{\boldsymbol{X}} \sim G_\varphi$. They then use $\phi'_\theta$ to filter $\hat{\boldsymbol{X}}$ for high-performing sequences, retrain $G_\varphi$, and iterate this process until convergence. This scheme is identical to the cross-entropy method with a VAE as the generative model, an important optimization scheme (Rubinstein, 1999). In follow-up work, termed Conditioning by Adaptive Sampling (CbAS) (Brookes et al., 2019), the authors aim to address the pitfall in which the oracle is biased, and gives poor estimates outside its training domain $\mathcal{D}$. The authors enforce a soft pessimism penalty for samples that are very distinct from those that the oracle could have possibly learned from. Specifically, they modify the paradigm so that as the generator updates its parameters $\varphi_t \to \varphi'_t$ while training on samples in the tail of the distribution, it discounts the weight of the samples $\boldsymbol{x}_i$ by $\frac{\Pr(\boldsymbol{x}_i|G;\varphi_0)}{\Pr(\boldsymbol{x}_i|G;\varphi_t)}$. In other words, if the generative model that was trained on the original data was more enthusiastic about a sample than the one that has updated according to the oracle's recommendations, that sample is up-weighted in the next training round (and vice versa). Notably, as the oracle is not updated during the process, there are two rounds of experiments where they maximize the potential gains from their oracle given what it already knows: a round to create the oracle, and a round to improve the generative model. While it is trivial to repeat the process for multiple rounds, the process can be improved by incorporating information about how

many rounds it will be used for. We use DbAS and CbAS as state of the art representatives in this class of regularized generative algorithms.

**Reinforcement Learning (RL).** RL algorithms are another method used to approach this problem. As these algorithms learn to perform tasks by experience, their success is often dependent on whether interactions with the environment are cheap or if there is a good simulator of the environment which they can practice on (Mnih et al., 2015). However, in our setting, good simulators often do not exist, and sampling the environment directly can be very expensive. As a result, recent approaches to this problem have built locally accurate simulators and used them to train an RL agent. With DyNA-PPO, Angermueller et al. (2020) achieve state-of-the-art results in sequence design tasks following this approach. They train a policy network based on Proximal Policy Optimization (PPO) (Schulman et al., 2017) by simulating $\phi$ through an ensemble of models $\phi''$. The models are trained on all measured data so far, and those that achieve a high $R^2$ score in cross-validation are selected as part of the ensemble. Additionally, to increase the diversity of proposed sequences, they add a penalty for proposing samples that are close to previously proposed samples. They compare their results against some of the methods mentioned above, as well as other RL training schemes, showing superior results. While DyNA-PPO reaches state of the art performance, a major drawback of policy gradient algorithms is that they are complex to implement, and their performance is highly sensitive to implementation (Engstrom et al., 2019). We use DyNA-PPO as a representative state of the art for all sequence design algorithms for our study.

Most recent sequence design algorithms do not use evolution-inspired methods. Despite their lack of popularity, as we show here, they are fairly strong baselines that should be considered. Here we investigate classic algorithms like standard Wright-Fisher process, without and without models, Covariance Matrix Adaptation evolutionary strategy, (CMA-ES) (Hansen & Ostermeier, 2001) and ADALEAD.

It's important to note that the focus of our study is distinct from those focused on better local approximate models (e.g. (Rao et al., 2019; Riesselman et al., 2017)), or semi-supervised and transfer learning approaches (e.g. (Madani et al., 2020; Biswas et al., 2020)).

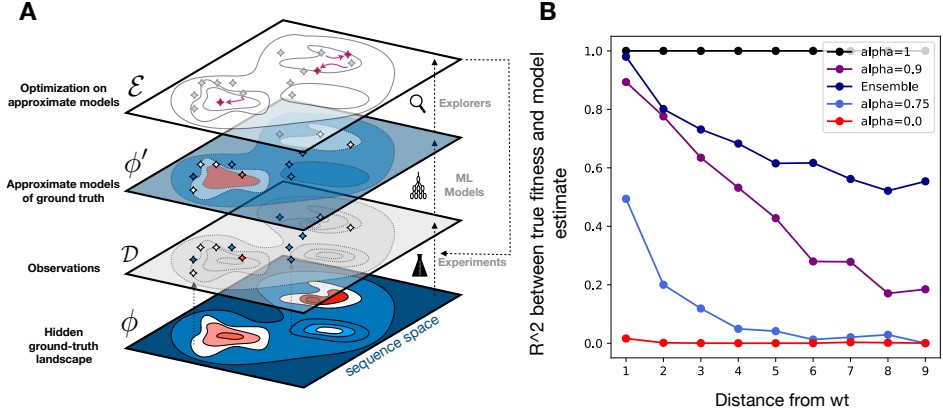

Figure 1: **A:** An overview of our scheme. An expensive-to-query **ground-truth oracle** $\phi : \boldsymbol{x} \to y$; a **local approximate model** $\phi'$ trained on samples $\mathcal{D}$ from $\phi$; an **exploration algorithm** $\mathcal{E}$ which is the primary interest of our study (e.g. ADALEAD, DynaPPO, . . . ). In our setting, the training of $\phi'$ is fully supervised. **B:** Noise corrupted abstract models $\phi'_\alpha$ allow for independent study of $\mathcal{E}$ on landscapes where ground truth can be simulated (here an RNA landscape of size 14, binding to one target). For comparison, the $R^2$ score for an ensemble of 3 CNNs trained on a random set of 100 sequences around the starting position is provided.

## 5 METHODS

In this study, we evaluate representative state of the art algorithms in addition to a simple greedy algorithm which we call Adapt-with-the-Leader (ADALEAD) (See Algorithm 1). The main advantage

of ADALEAD as a benchmark is that it is easy to implement and reproduce. The algorithm selects a set of seed sequences $\boldsymbol{x}$ such that $\phi(\boldsymbol{x})$ is within $(1 - \kappa)$ of the maximum $y$ observed in the previous batch. These seeds are then iteratively recombined and mutated (see appendix for details) and the ones that show improvement on their seed according to $\phi'$ are added to a set of candidates $M$. Finally, all candidates are sorted according to $\phi'$ and the top $B$ are proposed for the next batch. We consider the recombination rate $r$ and mutation rate $\mu$ as well as the threshold $\kappa$ as hyperparameters. However, the algorithm is fairly robust in performance for $\kappa, r < 0.5$. We note that recombination is known to be a powerful generative method which promotes diversity and avoids local maxima (Otwinowski & LaMont, 2019). Despite this, the performance gain due to recombination in ADALEAD is small (see supplement Fig. A2 for details).

---

**Algorithm 1** ADALEAD

---

**Input:** model $\phi'$, batch $b_t$, threshold $\kappa$, virtual evaluations $v$

Initialize parents $P \leftarrow \varnothing$
Initialize mutants $M \leftarrow \varnothing$
Update $\phi'$ with data from $b_t$
$\mathcal{S} = \{\boldsymbol{x} \mid \phi(\boldsymbol{x}) \geq \max_{y \in b_t} y \cdot (1 - \kappa), \forall \boldsymbol{x} \in b_t\}$
**while** $|M| < v \cdot |b_t|$ **do**
   $P = P \cup \text{RECOMBINE}(\mathcal{S})$
   **for** $\boldsymbol{x}_i \in P$ **do**
      $\{\boldsymbol{x}_{i1}, \ldots, \boldsymbol{x}_{ik}\} = \text{ROLLOUT}(\boldsymbol{x}_i, \phi')$
      $M = M \cup \{\boldsymbol{x}_{i1}, \ldots, \boldsymbol{x}_{ik}\}$
   **end for**
**end while**
Use $\phi'$ to select $b_{t+1}$, the top $|b_t|$ sequences from $M$
RETURN $b_{t+1}$

---

As ADALEAD perturbs the best (known) sequences, it is a greedy algorithm where the threshold $\kappa$ determines the greediness. However, it is adaptive in the sense that given a fixed threshold $\kappa$, when the optimisation surface is flat, many sequences will clear the $(1 - \kappa) \cdot \max_{y \in b_t} y$ filter, and therefore the algorithm encourages diversity. When the surface has a prominent peak, it will opt to climb rapidly from the best-known samples. As we will see, this yields a robust, yet surprisingly effective algorithm that uses the same principle of hill-climbing as a Wright-Fisher process, but faster and more scalable to compute (as it does not require fitness based sampling), and is supported by the intuition behind Fisher's fundamental theorem (see Otwinowski & LaMont (2019) for a helpful discussion). Notably, this is distinct from rank-based and quantile-based algorithms, where diversity may be compromised due to the dominance of sequences that are trivial changes to $\boldsymbol{x}_t^*$, and hence are more likely to remain at the same local optima. The ROLLOUT procedure mutates (with mutation rate $1/L$) proposed candidates in $\mathcal{S}$ until $\phi'(\boldsymbol{x}_{i,k}) < \phi'(\boldsymbol{x}_{i,0})$ where $k$ is the number of times a candidate has been subject to a mutation operation. Finally, all candidates are sorted according to $\phi'$ and the top $B$ are proposed for the next batch. We find that the rollout process has a small beneficial effect on the performance of the algorithm (see supplement Fig. A4). Overall, we speculate that ADALEAD takes advantage of correlation structure of biological sequences. Namely that sequences close to each other are more likely to have similar values. Since it always starts from the best **known** sequences, it ensures better robustness (less dependence on the model). The optimization side is greedy hill-climbing with the help of the model's foresight. We found it surprising that such a simple method can compete with more sophisticated state of the art approaches presented here.

In order to focus our attention on comparing the power of exploration algorithms, rather than the power of an oracle, we make use of, but do not limit ourselves to, an abstract model $\phi'_\alpha$, which is a noise-corrupted version of the ground truth landscape. Specifically,

$$\phi'_\alpha = \alpha^d \phi + (1 - \alpha^d)\epsilon \tag{1}$$

where $d$ is the distance from the closest measured neighbor and $\epsilon$ is a noise parameter sampled from an exponential distribution with $\lambda$ equal to $\phi$ operating on the closest measured neighbor[1]. We find

---

[1]An alternative approach is to let the noise be a random sample from the empirical distribution of known mutants; since $\epsilon$ behaves more stochastically in this setting, we do not evaluate the models with this approach.

that this setup allows us to emulate the performance of trained models well, while controlling for model accuracy. We also define the null model $\phi'_{\text{null}}$ as an exponential distribution with $\lambda$ equal to the average measured fitness (Orr, 2010). The null model is a special case of the abstract model where $\alpha = 0$. Importantly, this abstraction allows us to control $\alpha$ to investigate consistency and robustness (see supplement for additional description and Fig. 1B for a comparison of abstract and empirical models on RNA landscapes).

To validate the applicability of this strategy, we also use it with trainable sequence models, including a simple linear regressor, random forest regressors, Gaussian process regressors, and several neural network architectures. As we show using the abstract models, ADALEAD is consistent, and any improvement on the model is beneficial for the algorithm. Although ADALEAD is compatible with single models, we implement and recommend using it with an ensemble of models. Additionally, our implementation is adaptive: model estimates are re-weighted based on how well they predicted the labels on sequences proposed on the previous batch. Herein, we show the results from an ensemble of 3 CNN models as a representative choice of the empirical models, denoted by $\phi''$. This ensemble was our strongest empirical model across different landscapes, among those mentioned above.

## 6 EXPERIMENTS

Prior to discussing the empirical experiments, two preliminaries are noteworthy. Firstly, proposing a general algorithm that can optimize on arbitrary landscapes is not possible, and even local optima can take an exponential number of samples to find with hill-climbing algorithms (Wolpert et al., 1997; Kaznatcheev, 2019). Nonetheless, since biological landscapes are governed by physio-chemical laws that may constrain and structure them in specific ways, some generalizability can be expected. Secondly, the problem of choosing suitable optimization challenges that are biologically representative requires some consideration. Biological fitness landscapes are notoriously hard to model outside the locality where rich data is available (see Bank et al. (2016) as well as Fig. A3 in the supplement). As such, models that are built on data around some natural sequence are often only representative in that context, and the model becomes pathological rapidly as the distance from the context increases. This is the motivation for Brookes et al. (2019) as an improvement on Brookes & Listgarten (2018) to ensure the model $\phi'$ is not trusted outside its zone of applicability. Here, we are interested in exploring deep into the sequence space, through multiple "experimental design" rounds. Hence it is highly preferable that the quality of our ground-truth simulator $\phi$ remains consistent as we drift away from the starting sequence. Note that this is not the case for models that are trained on empirical data such as those in Rao et al. (2019) where the model is accurate in local contexts where data is available, but the behavior outside the trust region is unknown and may be pathological.

We test our algorithms on multiple sequence design tasks. We first choose contexts in which the ground truth and the optimal solutions are known (to the evaluator). We then challenge the algorithms with more complex landscapes, in which the ground truth may be queried, but the optimal solution is unknown.

**TF binding landscapes**. Barrera et al. (2016) surveyed the binding affinity of more than one hundred transcription factors (TF) to all possible DNA sequences of length 8. Since the oracle $\phi$ is entirely characterized, and biological, it is a relevant benchmark for our purpose (Angermueller et al., 2020). We randomly sample five of these measured landscapes, and evaluate the performance of these algorithms on these landscapes. For each landscape, we start the process at 13 random initial sequences which are then fixed. We do not pre-train the algorithms on other landscapes before using them, since TF binding sites for different proteins can be correlated (e.g. high-performing sequences in some landscapes may bind many proteins well). Should pre-training be required, we impose a cost for collecting the data required. We shift the function distribution such that $y \in [0, 1]$, and therefore $y^* = 1$. We show the results of our optimization tasks on these landscape in Fig. 2A. All algorithms perform similarly well in terms of optimization in these landscapes, suggesting that while the task itself is biologically suitable, the landscape is rather easy to optimize on, partially due to its size.

**RNA landscapes**. The task of predicting RNA secondary structures has been well-studied and is relatively mature. Classic algorithms use dynamic programming to efficiently fold RNA structures. RNA landscapes are particularly suitable because they provide a relatively accurate model of biological sequences that is consistent across the domain, even for landscapes of size $4^{100}$, and are

| Landscape | $y_\tau$ | $\phi'$ | # peaks | AdaLead | DynaPPO | CbAS/DbAS | CMAES |
|---|---|---|---|---|---|---|---|
| 5 TF binding | $> 0.75$ | $\alpha = 1$ | 67.6 | **31.29** $\pm 9.24$ | 22.85 $\pm 11.05$ | 2.57 $\pm 1.08$ | 26.16 $\pm 10.74$ |
| 5 TF binding | | ensemble | | **11.65** $\pm 4.12$ | 4.09 $\pm 4.23$ | 1.79 $\pm 1.25$ | 10.02 $\pm 3.6$ |
| 5 TF binding | | $\alpha = 0$ | | **5.02** $\pm 2.42$ | 1.76 $\pm 1.54$ | 1.99 $\pm 1.64$ | 2.1 $\pm 1.92$ |
| 5 TF binding | $> 0.9$ | $\alpha = 1$ | 22.6 | **21.87** $\pm 5.92$ | 13.33 $\pm 6.04$ | 1.35 $\pm 1.08$ | 15.05 $\pm 4.59$ |
| 5 TF binding | | ensemble | | **9.56** $\pm 3.7$ | 1.55 $\pm 1.9$ | 0.79 $\pm 0.9$ | 7.53 $\pm 3.4$ |
| 5 TF binding | | $\alpha = 0$ | | **3.7** $\pm 1.93$ | 0.57 $\pm 0.88$ | 0.9 $\pm 0.9$ | 0.9 $\pm 1.07$ |
| 5 TF binding | $= 1$ | $\alpha = 1$ | 1 | **0.97** $\pm 0.16$ | 0.53 $\pm 0.5$ | 0.05 $\pm 0.21$ | 0.67 $\pm 0.47$ |
| 5 TF binding | | ensemble | | **0.31** $\pm 0.46$ | 0.08 $\pm 0.27$ | 0.03 $\pm 0.17$ | **0.31** $\pm 0.46$ |
| 5 TF binding | | $\alpha = 0$ | | **0.21** $\pm 0.36$ | 0.02 $\pm 0.11$ | 0.06 $\pm 0.22$ | 0.04 $\pm 0.18$ |
| RNA14_B1 | $> 0.75$ | $\alpha = 1$ | 353 | **12.8** $\pm 5.26$ | 9.2 $\pm 10.8$ | 1.0 $\pm 0.7$ | 1.4 $\pm 0.5$ |
| RNA14_B1 | | ensemble | | **3.8** $\pm 2.16$ | 1.6 $\pm 2.5$ | 1.0 $\pm 0.8$ | 0.67 $\pm 1.0$ |
| RNA14_B1 | | $\alpha = 0$ | | **0.5** $\pm 0.52$ | 0.3 $\pm 0.67$ | 0.2 $\pm 0.63$ | 0 |
| RNA14_B1 | $> 0.9$ | $\alpha = 1$ | 37 | **3.6** $\pm 1.67$ | 1.4 $\pm 2.07$ | 0.4 $\pm 0.25$ | 0.4 $\pm 0.7$ |
| RNA14_B1 | | ensemble | | **1.2** $\pm 0.83$ | 0 | 0.33 $\pm 0.58$ | 0.4 $\pm 0.54$ |
| RNA14_B1 | | $\alpha = 0$ | | **0.2** $\pm 0.42$ | 0 | 0 | 0 |
| RNA14_B1 | $= 1$ | $\alpha = 1$ | 3 | **0.4** $\pm 0.54$ | 0 | 0.2 $\pm 0.4$ | 0 |
| RNA14_B1 | | ensemble | | **0.2** $\pm 0.44$ | 0 | 0 | 0 |
| RNA14_B1 | | $\alpha = 0$ | | **0.1** $\pm 0.31$ | 0 | 0 | 0 |
| RNA14_B1+2 | $> 0.75$ | $\alpha = 1$ | 33 | **4.8** $\pm 0.83$ | 1.8 $\pm 0.83$ | 0.4 $\pm 0.54$ | 0.2 $\pm 0.44$ |
| RNA14_B1+2 | | ensemble | | **2.6** $\pm 1.34$ | 0.0 | 0.0 | 0.0 |
| RNA14_B1+2 | | $\alpha = 0$ | | **0.3** $\pm 0.67$ | 0.1 $\pm 0.31$ | 0 | 0 |
| RNA14_B1+2 | $> 0.9$ | $\alpha = 1$ | 9 | **1.4** $\pm 0.89$ | 0 | 0 | 0.4 |
| RNA14_B1+2 | | ensemble | | **1.2** $\pm 1.09$ | 0 | 0 | 0 |
| RNA14_B1+2 | | $\alpha = 0$ | | **0.2** $\pm 0.31$ | 0 | 0 | 0 |
| RNA14_B1+2 | $= 1$ | $\alpha = 1$ | 3 | **0.6** $\pm 0.54$ | 0 | 0 | 0 |
| RNA14_B1+2 | | ensemble | | **0.2** $\pm 0.44$ | 0 | 0 | 0 |
| RNA14_B1+2 | | $\alpha = 0$ | | **0.1** $\pm 0.31$ | 0 | 0 | 0 |

Table 1: We compare the algorithms based on the number of optima above a certain threshold $y_\tau$ each of them finds. The total number of optima has been computed by brute force in each landscape. The algorithms were run with an empirical ensemble model, a perfect model ($\alpha = 1$) and an uninformative model ($\alpha = 0$), querying the ground truth with a total of 1000 sequences, and the surrogate model with ratio of $v = 20$ and at least 5 initializations each (mean and standard deviation reported). For TF landscapes (size $4^8$ each), we average the scores across 5 landscapes (each with 13 initiations). AdaLead finds high-performing peaks more consistently than the other algorithms. It regularly finds the global optimum in an RNA binding landscape landscape of size $4^{14}$, with 2925 local peaks in total (RNA14_B1). It also outperforms other algorithms in a similarly sized composite landscape with two binding targets (RNA14_B1+2), with 806 peaks overall.

non-convex with many local optima (see Fig. A3). We use the ViennaRNA package to simulate binding landscapes of RNA sequences as $\phi$ (Lorenz et al., 2011).

We test our algorithms on many RNA binding landscapes of various complexity and size (e.g. sequence length 14–100). In short, AdaLead outperforms the other algorithms in all of the attempted landscapes. As a basic ground-truth test, we optimize sequences of size 14 for binding hidden 50 nucleotide targets. We use the duplex energy of the sequence and the target as our objective $y$, which means that the sequence can bind the target in multiple different ways, producing many local minima. We also consider more complex landscapes with hidden targets and compute the objective as $\sqrt{y_1 y_2}$. Due to the size of these landscapes we can enumerate all $4^{14}$ sequences with the oracle, and as with the TF binding landscapes, find out how well the algorithms explore the space. Table 1 summarizes the number of local optima with function greater than $y_\tau$ that each algorithm finds. We define local optima in this case as sequences whose immediate neighbors all have lower $y$.

We also test RNA sequences of size 100, that bind hidden targets of size 100. In these cases we do not know the actual optima in advance, hence, we estimate by computing the binding energy of the perfect complement of the target. Using this normalization, $y^* \approx 1$. Like before, we use both single

and double target binding objectives. Additionally, we define conserved patterns in sequences which would not allow mutations, meaning the sequence needs to preserve those positions in order to remain viable. In these cases, we conserve a fifth of the sequence, which results in roughly a fifth of the landscape providing no gradients. This resembles the scenario for many biological objectives, where gradients are not available over large regions of the space. We refer to this challenging landscape as "Swampland" (see a breakdown in Fig. A3). Please refer to section A3 for a more detailed discussion. Despite using a greedy heuristic, ADALEAD outperforms the other algorithms in landscapes that

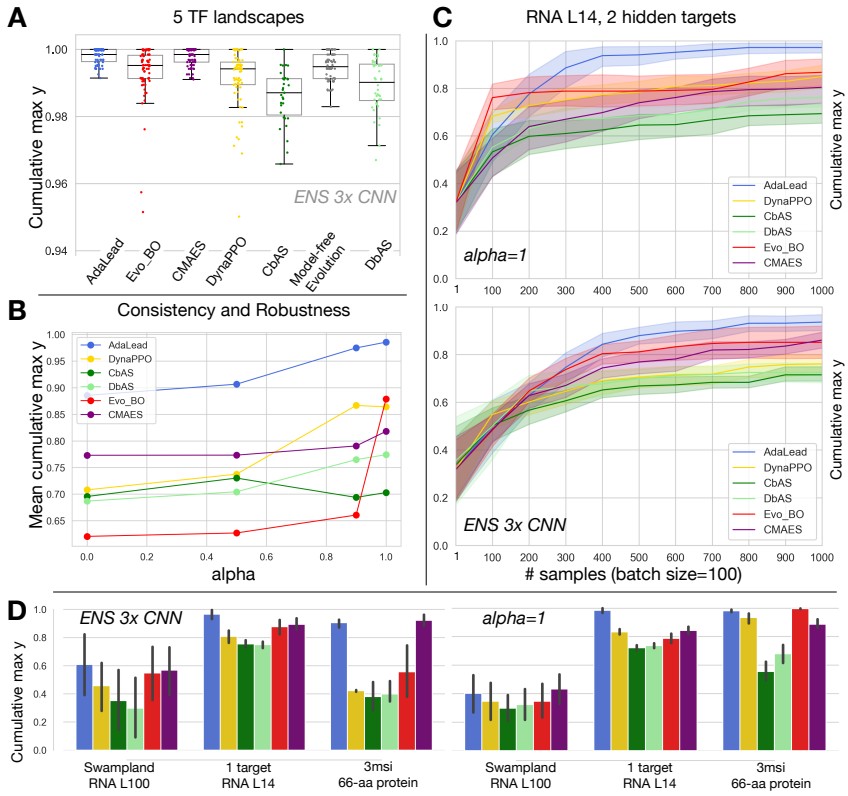

Figure 2: We record the cumulative maximum over all sequences generated by each algorithm when run with 10 batches of size 100, and $v = 20$. All scores on the y-axis normalized to known or estimated maximum possible value. **A:** The cumulative maximum achieved by each algorithm on TF binding landscapes (13 initializations), using an ensemble of 3 CNNs as the oracle ($\phi''$). On this simple landscape, even a model free evolution algorithm can optimize well. **B:** Consistency (performance vs. model quality $\alpha$) and robustness (performance at low $\alpha$) of the algorithms on a 2 target RNA landscapes of $L = 14$. **C:** Time evolution of the cumulative maximum over an RNA landscape with sequence length 14, and 2 hidden targets (5 initializations, $\alpha = 1$). Top: $\phi_{\alpha=1}$, bottom: $\phi''$, ensemble of 3 CNNs. **D:** Comparison of overall performance for 3 landscape classes. Swampland landscapes show high variance due to the difficulty of finding good sequences starting from dead sequences. ADALEAD, and evolutionary algorithms in general tend to be strongly competitive in more complex landscapes.

are highly epistatic (include a lot of local peaks). As shown in Fig. A3, even the set of shortest path permutations on the landscapes between one of the starting positions and the global peak may include valleys of multiple deleterious steps. The time evolution of the best sequence found at each batch, shown in Fig. 2C, suggests that some algorithms are faster to climb in the first couple of batches, but none outperform ADALEAD in the longer horizon. As we show in Fig. 2B, ADALEAD is also more robust (performs well even with an uninformative model), and as consistent as all the other algorithms. The relative ranking of algorithms remains similar to the $\alpha = 1$ case when CNN ensembles are used (Fig. 2C).

**Protein design**. As a final challenge we also compare the performance of algorithms with multiple protein design tasks. While ground-truth simulators for protein design are much less accurate than the RNA landscapes, the larger alphabet size of $\sim 20$ and complexity of the landscapes are of high relevance. In this case we use PyRosetta (Chaudhury et al., 2010) as $\phi$. The Rosetta design objective function is a scaled estimate of the folding energy, which has been found to be an indicator of the probability that a sequence will fold to the desired structure (Kuhlman et al., 2003). We optimize for the structure of 3MSI, a 66 amino acid antifreeze protein found in the ocean pout (DeLuca et al., 1998) starting from 5 sequences with 3–27 mutations from the wildtype. Here, we normalize energy scores by scaling and shifting their distribution and then applying the sigmoid function.

## 7 CONCLUSIONS AND FUTURE DIRECTIONS

We implement an open-source simulation environment FLEXS that can emulate complex biological landscapes and can be readily used for training and evaluating sequence design problems. We also include "Swampland" landscapes with large areas of no gradient, a biological aspect of sequence design rarely explored (see Fig A3). We also provide additional interfaces for protein design based on trained black-box oracles (e.g. (Rao et al., 2019) that we don't study for reasons explained in the manuscript. Additionally, we proposed a simple evolutionary algorithm, ADALEAD as a competitive benchmark. We demonstrate that ADALEAD is able to robustly optimize in challenging settings, and consistently performs better as model performance improves. We show that in general, simple evolutionary algorithms are strong benchmarks to compete against. While we have investigated consistency and robustness for the queries to ground truth oracle, the same concepts also apply to variations in $v$. These would affect sample efficiency, and scalability of the algorithms. There are also other properties of interest, also mentioned in Purohit et al. (2018) (e.g. *independence*), which are closely connected to consistency and robustness, where the algorithm can operate with oracles with different biases and noise profiles. Additionally, in the online batch setting, for any fixed total sequences proposed, the algorithm is expected to pay a performance penalty as the batch size grows. This is due to lack of model updates for the sequences proposed within each batch. Algorithms that incur lower penalties can be desirable in low-round batch setting. This is known as *adaptivity*. We do not evaluate these properties directly in this work, but implement tools that allow for their study within FLEXS.

We hope that FLEXS provides a useful environment for future development of better sequence design algorithms, and hope that ADALEAD help discipline such efforts towards more simple approaches that are reproducible and translatable in practice.

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

## A    APPENDIX

### A.1    NOISE MODELS

#### A.1.1    NOISY ORACLE

In order to control for model accuracy, we define noise-corrupted versions of the landscape oracle

$$\phi'_\alpha = \alpha^d \phi + (1 - \alpha^d)\epsilon,$$

where $d$ is the distance from the closest measured neighbor and $\epsilon$ is a noise parameter sampled from an exponential distribution with the rate parameter $\lambda$ equal to $\phi$ operating on the closest measured neighbor. An alternative approach is to sample $\epsilon$ from the empirical distribution of known mutants; since $\epsilon$ behaves more stochastically in this setting, we use the former approach, while we confirm that the results are qualitatively the same if the second approach is used.

In Fig. 1B we show the performance of these models on an RNA14_B1 landscape as $\alpha$ is varied between 0 and 1. We also show the predictive power of the empirical CNN ensembles, when trained on 100 random mutants with varying distances from the wild type on the same landscape. The noise-corrupted oracles allow testing for consistency and robustness in particular.

#### A.1.2    EMPIRICAL MODELS

For our sandbox, we implement a suite of empirical models. We use linear regression, random forest regression from the `scikit-learn` library (Pedregosa et al., 2011). We also implement two neural architectures: (1) A "global epistasis" network that first collapses the input to a single linear unit, and then a non-linear transform through two 50 unit ReLUs, and a final single linear unit (Otwinowski et al., 2018). (2) A convolutional neural network. We tested a variety of other architectures and `sklearn` models, but found these sufficient as representative models. We find that the CNN is often the most accurate model. For ensembling, we use ensembles of 3 initializations of the CNN architecture. Ensembles may be run in "adaptive mode" where the model outputs are averaged based on weights proportional to the $R^2$ of the model on the data generated in the previous step. While ensembles come with the additional benefit of allowing uncertainty estimation (which is useful in most of our algorithms), the performance gain of using them is often small. In the paper, we show the results for an adaptive ensemble of 3 CNNs, which was the strongest empirical model we attempted.

### A.2    ALGORITHMS

#### A.2.1    BAYESIAN OPTIMIZATION

We first test classical GP-BO algorithms on TF landscapes, as they are of small enough ($4^8$ total sequences) to enumerate fully. We use a Gaussian process regressor (GPR) with default settings from the `sklearn` library. Furthermore, we lift the virtual screen restriction for this particular optimization method. We propose a batch of sequences as determined by an acquisition function. We use the standard EI, UCB as well as Thompson sampling (where the posterior is sampled for each sequence, and top $B$ are selected for next round). The GPR based, enumerative approach scales poorly and cannot accommodate domains where the sequence space is much larger (e.g. RNA landscapes or protein landscapes).

As a compromise, we also implement a BO-guided evolutionary algorithm, where mutants are generated at random from sequences, defining a tractable action space. To generate the batch sequences in each round, we take a state sequence $s$ and sample $v$ sequences with per position mutation probability of $\frac{1}{L}$. Instead of GPRs, we use ensembles of the empirical models to compute uncertainty in the posterior, similar to (Belanger et al., 2019). We evaluate the ensemble on each proposed mutant $s'_{i,0}$, and use EI and UCB acquisition functions for selecting $s'_{i,0}$ to mutate further to $s'_{i,1}$. We stop adding mutations once $\text{Var}(\phi''(s_{i,t})) > 2 \cdot \text{Var}(\phi''(s_{i,0}))$, or once we reach the virtual screen limit. We collect all candidates that were generated through this process and we then use Thompson sampling based on $\phi''(s)$ to choose a subset of size $B$ as our batch. After the batch is filled and the ensemble is updated, the state sequence for the next batch is chosen to be the sequence from the previous batch with the highest predicted score (Frazier, 2018). As shown in Fig. A1A, the evolutionary BO is competitive with, or better than, the enumerative BO algorithms we attempted on the TF landscapes. We therefore use the evolutionary BO in the main paper.

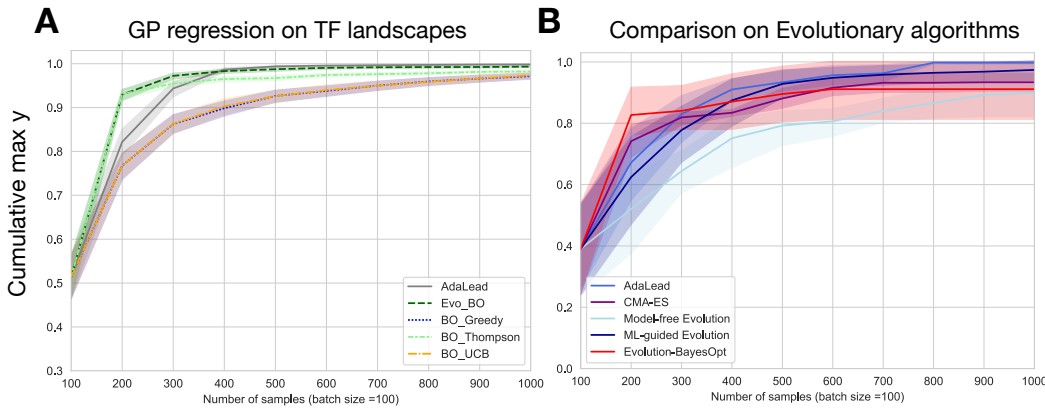

Figure A1: **A:** Running GP-based BO on the TF binding landscape with *full* enumeration of the sequence space ($v = \infty$). For comparison, we show that the evolutionary BO, used in the paper as a benchmark, outperforms these methods. **B:** A comparison of multiple evolutionary algorithms that were run on RNA binding landscapes of length 14 and one target, with similar $\mu = 1/L, r = 0.2$.

### A.2.2  ADALEAD

We performed basic parameter tuning for ADALEAD, testing recombination rates, mutation rates and threshold. For comparisons with other algorithms, we use recombination rate of $0.2$ (i.e. $1/5$ probability of a crossover at each position), mutation rate of $1/L$, where $L$ is the sequence size, and threshold of $\tau = 0.05$. We find that presence of recombination helps the performance of the algorithm both in optimality and ability to find diverse peaks. However, both of these effects are small (see Fig. A2) and most of the benefits are present within rate $< 0.3$, above which the stochastic effects tend to be detrimental with noisier models. While higher mutations rates can help with exploration we chose $1/L$ to accommodate the simpler models and uniformity across all algorithms (some that introduce one change at a time). The $\tau$ parameter begins to be effective below $0.5$, but tends to be too restrictive (resulting in less than enough seeds when generating) when $\tau < 0.05$. We use $\tau = 0.05$ in shown experiments. Overall, ADALEAD is fairly robust to these parameters. Since ADALEAD is a evolutionary algorithm, we also compare its performance to other evolutionary algorithms as benchmarks. Our results, shown in Fig. A1B, show that ADALEAD is roughly equivalent to a model-guided Wright-Fisher process (this is expected, as ADALEAD operates on the same hill-climbing principle although it is faster to compute and less memory intensive, and hence it can be scaled better). It consistently outperforms model-free evolution (WF), CMA-ES, and Evolutionary BO.

### A.2.3  DBAS

We implement the sampling algorithm introduced in Brookes & Listgarten (2018) with a variational autoencoder (VAE) (Kingma & Welling, 2013) as the generator. The encoder and decoder of the VAE both consist of three dense layers with 250 exponential linear units. There is a 30% dropout layer after the first encoder and the second decoder layer as well as batch normalization after the second encoder layer. The input is a one-hot-encoding of a sequence and the output layer has sigmoid neurons that result into a positional weight matrix as a reconstruction of the input. The latent space dimension is 2. We use the Adam optimizer (Kingma & Ba, 2014). In each cycle, the DbAS algorithm starts by training the VAE on the sequences whose fitness is known and selects the top 20% as scored by the oracle. New samples are then generated by drawing from the multivariate Gaussian distribution of the latent space of the VAE. Once again, the top 20% of the generated sequences (or the ones whose fitness is above the 80th percentile of the previous batch, whichever is the stricter threshold) are selected and the process is repeated until convergence (which in our case is defined as not having improved the top scoring sequence for 10 consecutive rounds) or the computational budget is exhausted. At that point the batch of sequences proposed in the latest iteration is returned.

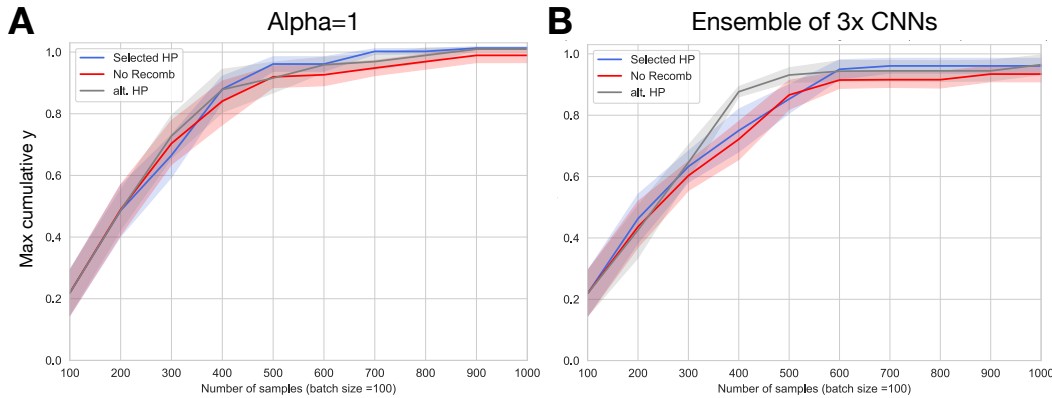

Figure A2: Effects of these hyperparameters on the performance of ADALEAD, on RNA landscape of length 14 and two hidden targets. ADALEAD is robust to hyperparameter choices for $\kappa, r$. The setting used in the paper shown is in blue ($r = 0.2, \kappa = 0.05$). The same $\kappa$ with no recombination is shown in red. All other hyperparameters $r \in [0, 5]$ and $\kappa$ are shown as "alt. HP". $\mu$ was set to mirror the $\mu$ used for all other algorithms, and $v = 20$. **A:** The case where $\alpha = 1$, i.e. the model has perfect information. **B:** When an ensemble of 3 CNNs was used.

### A.2.4 CBAS

We implement the adaptive sampling algorithm proposed in Brookes et al. (2019). The parameters and the generator are identical to the DbAS implementation, see Section A.2.3. The difference here is that, compared to a DbAS cycle, where the top sequences in each cycle are selected based on the fitness predicted by the oracle, in a CbAS cycle the sequences are weighted by the score (in this case the reconstruction probability of the VAE) assigned to it by the generator trained on the ground truth data divided by the score assigned to it by the generator trained on the all sequences proposed so far.

### A.2.5 CMA-ES

We adapt the covariance matrix adaptation evolution strategy (CMA-ES) (Hansen & Ostermeier, 2001) for the purpose of sequence generation. Let $n = A \times L$ be the product of the alphabet length and the sequence length. We initialize the mean value $\mathbf{m} \in \mathbb{R}^n$ of the search distribution to a zero vector, and the covariance matrix $\mathbf{C} \in \mathbb{R}^{n \times n}$ to an identity matrix of the same dimension.

At every iteration, we propose $\lambda$ sequences, where $\lambda$ is equal to the batch size. We sample $\lambda \times V$ sequences, where $V$ is the virtual screening ratio. Every sample $\mathbf{x} \sim \mathcal{N}(\mathbf{m}, \mathbf{C})$ is converted into a one-hot representation of a sequence $\mathbf{x}$ by computing the argmax at each sequence position. A model provides a fitness value for the proposed sequence. Out of the $\lambda \times V$ proposed samples, the top $\lambda$ sequences (by fitness value) are returned to be evaluated by the oracle. Because the sampled sequences are continuous but the one-hot representations are discrete, we normalize $\mathbf{m}$ such that $\|\mathbf{m}\|_2 = 1$ to prevent value explosion.

### A.2.6 PPO

As a sanity check for ensuring that DyNA-PPO is implemented well, we also test proximal policy optimization (PPO) to train the policy network which selects the best action given a state. The policy network is pretrained on $\frac{B \times V}{2}$ sequences, where $B$ is the batch size and $V$ is the virtual screening ratio. The remaining budget for evaluating sequences is then amortized among the remaining iterations – that is, each sequence proposal step trades some of its evaluation power in order to train a stronger policy network in the beginning. In each iteration where we propose sequences, we allow the agent to propose $B \times V$ sequences, and take the top $B$ sequences according to their fitnesses predicted by the model.

We use the TF-Agents library (Guadarrama et al., 2018) to implement the PPO algorithm. Our policy and value networks each consist of one fully-connected layer with 128 hidden units. Our results are consistent with those in Angermueller et al. (2020).

### A.2.7    DyNA-PPO

We closely follow the algorithm presented in Angermueller et al. (2020). We perform 10 experiment rounds. In each experiment round, the policy network is trained on samples collected from an oracle. Each model comprising an ensemble model is then trained on this data. The top models are retained in the ensemble, while the remaining models are removed. Initially, the ensemble model is composed of models with the same `sklearn` architectures as in Angermueller et al. (2020) as well as several neural network architectures which we add. We compare models based on their cross-validation score; top models cross a predetermined threshold (which we also set to be 0.5). We compute the cross-validation score via five-fold cross-validation scored on the same section for each model.

The ensemble model serves to approximate the oracle function. The mean prediction of all models in the ensemble is used to approximate the true fitness of a sequence. For each experiment round, we perform up to 20 virtual rounds of model-based optimization on each sequence based on the outputs of this ensemble model. A model-based round is ended early if the ensemble model uncertainty (measured by standard deviation of individual model rewards) is over twice as high as the uncertainty of the ensemble in the first model-based evaluation.

As mentioned in the main text, TRPO algorithms can be sensitive to implementation (Engstrom et al., 2019). To ensure that we build a fair comparison, we implement a variant of DyNA-PPO that uses a sequence generation process akin to evolutionary algorithms (which we call mutative), as well as one that follows closely to that of the paper (which we call constructive). In the first case, when proposing a new sequence, the agent will begin at a previously measured sequence and mutate individual residues until the reward (which is the fitness value of a sequence) is no longer increasing, or until the same move is made twice (signalling that the agent thinks that no better action can be taken). In the constructive case in Angermueller et al. (2020), the agent will add residues onto an originally empty sequence until it reaches the desired sequence length, which is fixed beforehand. In this case, the step reward is zero until the last step is reached, in which case it is the fitness value of the sequence. We find that the mutative version of the algorithm performs better than the constructive version, likely due to the rewards no longer being sparse in the mutative setting.

Additionally, the DyNA-PPO algorithm as presented in Angermueller et al. (2020) trains the policy on a set of true fitnesses of sequences before entering the proposal loop. In our setting, all explorers are allowed to make $B$ queries to assess the true fitness of sequences, equal to the batch size of sequences proposed at the end of a round. We further limit the computational queries to $v \times B$ samples (a difference with the original algorithm). No hyper-parameter tuning on held-out landscapes are done, as opposed to the original paper.

### A.3    Landscapes

### A.3.1    RNA Landscapes

To better clarify the structure of the RNA landscapes, and demonstrate their complexity, particularly of the Swampland fitness landscapes, we break down the components that go into them in Fig. A3. Even small landscapes of size 14 show this type of non-convexity that is seen in the figure. We pick 6 sequences of interest: The max peak for hidden target 1, the max peak for hidden target 2, the starting position (termed wildtype), the top known sequence in the swampland landscape and the top sequence found by a model-free WF process. We then compute 30 direct (shortest mutational paths) between each pair of these sequences, and show their fitness on the landscape. It is clear that there is significant non-monotonicity and permutations of order of mutations can change the shape of the trajectory. We believe that these landscapes enable an appropriate challenge for sequence design algorithms, while enabling full information of ground-truth (which can be queried quickly) and is consistent across the domain.

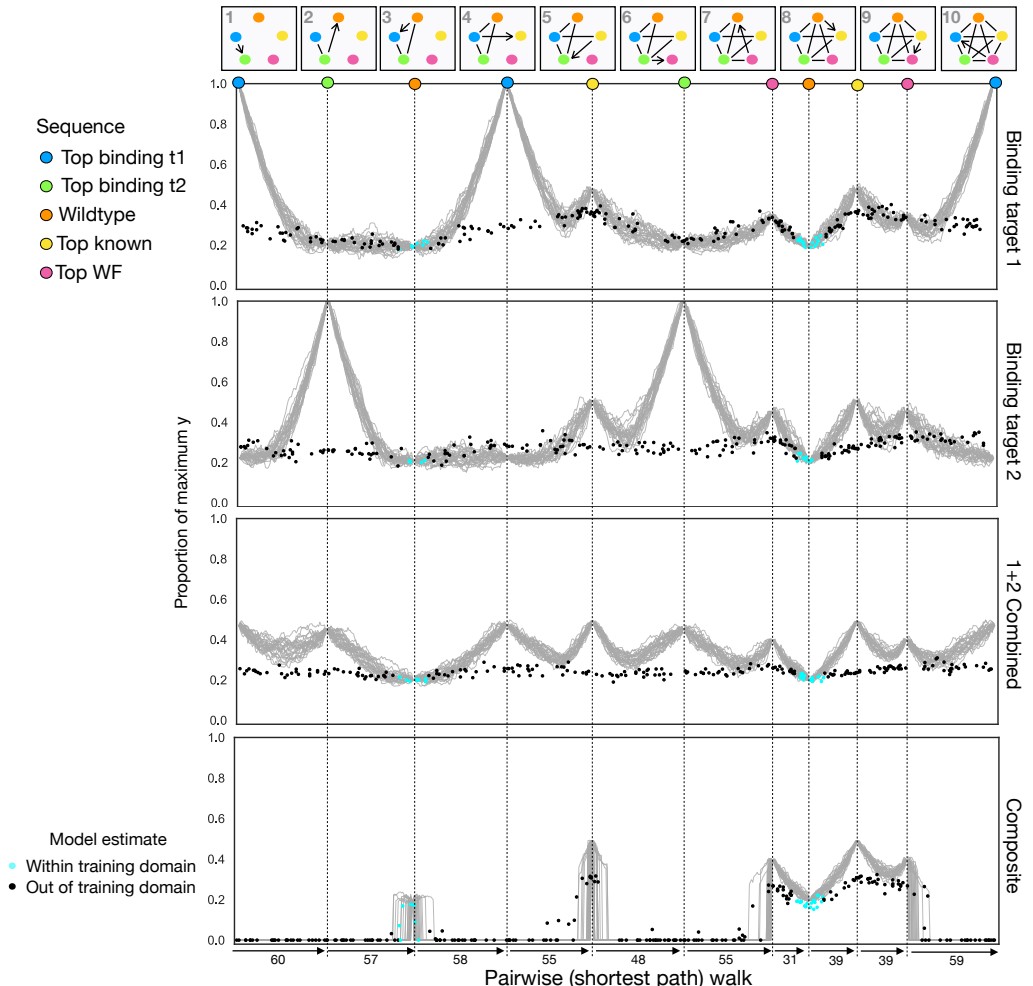

Figure A3: **A tour of a composite "Swampland" fitness landscape with sequence size 100 by direct walks between sequences of interest**. Colored circles represent sequences of significance. Each grey line is the fitness of a walk from one sequence to another. Walks are defined as shortest paths between two sequences, and different walks between the same sequences represent different orders of making the same set of substitutions (30 walks shown for each pair). The x-axis shows the number of steps between two sequences. The third panel shows the combined binding landscape of the first two panels (computed as $\sqrt{y_1 y_2}$). The composite "Swampland" landscape has the same targets as the combined landscape, but is also subject to the constraint that $20/100$ nucleotides cannot be mutated. We also train a CNN on 1000 mutants around the wildtype. The points underneath the plots represent the CNN's prediction of random samples from the paths. Cyan points show predictions within the same distance as the training set (here of max distance 12), and black points are extrapolations outside that range. The model fits the in-domain samples with high accuracy ($R^2 > 0.7$), but often misses global structure, as can be seen with the black points.

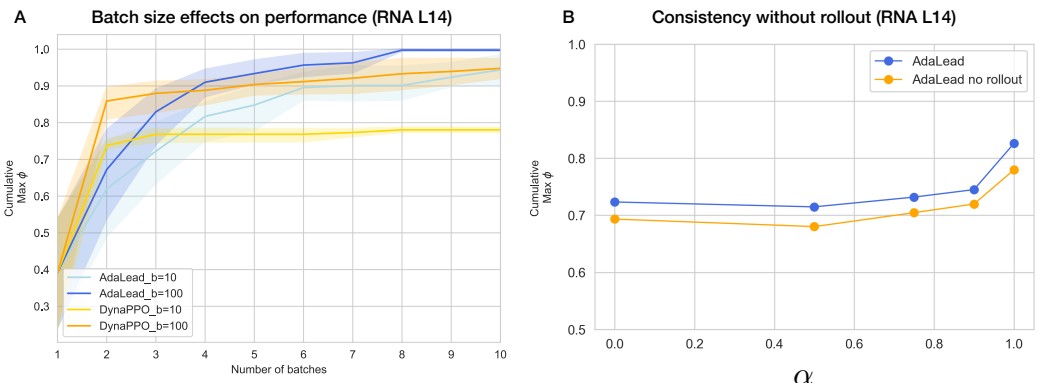

Figure A4: **A:** Effects of batch-size on ADALEAD, on RNA landscape of length 14 and two hidden targets.The case where $\alpha = 1$, i.e. the model has perfect information. **B:** Ablation study for Adalead without ROLLOUT on the same landscape.

