# OpenReview forum: "AdaLead: A simple and robust adaptive greedy search algorithm for sequence design"
_ICLR.cc/2021/Conference — Reject_

### Official Review · AnonReviewer2 · 2020-10-25
**The authors introduce a new algorithm AdaLead to solve the problem of efficient design of biological sequences and FLEXS, an open-source simulation environment for sequence design.**

**Rating:** 3
**Confidence:** 3

**Review:**

The authors introduce a new algorithm AdaLead to solve the problem of efficient design of biological sequences and FLEXS, an open-source simulation environment for sequence design.

The structure of this paper is somewhat unconventional, the paper would benefit from more rigor and structure. There are 4 sections, the method section which is the section that contains the novel part (section 2) is a fairly short section (less than 2 pages). The experimental setting is crammed together with the results. So, the details of the outcome of the experiments are lost in the various implementation details. The experimental setting can be moved in a section where the authors talk about the benchmarks alone. Or perhaps in the appendix.  The authors talk about the contributions at the very end of the paper. A reader wants to know what the contributions of the paper are in the introduction section. It wasn't very clear for example that FLEXS was the main contribution of this paper until the very last section. FLEXS was barely mentioned at the beginning of the paper.

The second contribution of the paper according to the authors is the novel algorithm AdaLead. This evolutionary algorithm is poorly explained. How are the recombine and mutate operators defined for example? In addition, this algorithm seems simplistic and I'm not sure we can call this algorithm a novel contribution. The authors should make more explicit why is this evolutionary algorithm a novelty, how does that improve wrt EAs?

The authors dismiss the use of Bayesian optimization by saying that BO doesn't perform well on high-dimensional space and citing a tutorial by Frazier 2018 that says that usually BO is used on problems that have less than 20 dimensions. I'm not entirely sure why this is a problem for this work where Table 1 shows that the applications considered have a few variables only, up to 14 if I understand correctly. In addition, some BO frameworks such as SMAC (https://www.automl.org/automated-algorithm-design/algorithm-configuration/smac/)  were used on problems with more than 100 variables.

The authors compare against BO by implementing a BO algorithm based on EI and an evolutionary sequence generator. While it is not clear what this means, this is surely an uncommon BO algorithm. Why didn't the authors use one of the standard BO packages available? The experimental results are unsatisfactory without a proper BO comparison.

Finally, I question the interest of the ICLR audience for this type of study. While the paper would follow in the track: "applications in audio, speech, robotics, neuroscience, computational biology, or any other field" I found the algorithm being a specific application of a simplistic search algorithm to the domain of sequence design. What are the lessons learned? Is there any general insight that became available from this study? The improvement wrt to SOTA is shaky.

Minor remarks:
- Figure 2 is not cited in the text.
- CMA-ES is not introduced - even if the CMA-ES is a popular algorithm it should be properly introduced.
- Model-free evolution in Figure 2 is not introduced. What is that algorithm about?
- Figure 2, I'm not sure what is the y-axis in the performance plot. In these types of papers, we usually show simple regret or similar performance quantities. It seems that the y-axis is the higher the better and that 1 is the max achievable?
- Figure 2 should give a better understanding of what the figure means and what are the methods that perform best by writing about that in the text of the paper.
- Not clear to me why the queries to the surrogate model are sample-restricted. This is an uncommon setting in Bayesian optimization where usually the budget is restricted for the evaluation of the latent black-box function.
- Not entirely clear what the role of the cardinality |S| in the optimization is. Is this an optimization with unknown feasibility constraints problem formulation?

---

> ### Author Response · Authors · 2020-11-15
> **Major comments response**
>
> Thank you for pointing out various places in which the presentation of our paper could be improved. We will make special note of these areas in our preparation of the following manuscript.
>
> **“A reader wants to know what the contributions of the paper are in the introduction section. It wasn't very clear for example that FLEXS was the main contribution of this paper until the very last section. FLEXS was barely mentioned at the beginning of the paper.”**
>
> The primary contribution of our paper, in our view, is to build an environment that enables disciplined benchmarking on relevant biological challenges. As a component of the benchmarking environment, we show that a simple algorithm (AdaLead) is more robust and performant than SOTA. This was an unexpected ‘discovery’ when we implemented AdaLead as a simple baseline.  As defending the claim that AdaLead is a strong benchmark in the aspects that matter to this problem requires more effort, we have allocated a large proportion of the paper to this second task. We are more than happy to restructure the paper to make this clear.
>
> **“The authors dismiss the use of Bayesian optimization by saying that BO doesn't perform well on high-dimensional space and citing a tutorial by Frazier 2018 that says that usually BO is used on problems that have less than 20 dimensions. I'm not entirely sure why this is a problem for this work where Table 1 shows that the applications considered have a few variables only, up to 14 if I understand correctly. In addition, some BO frameworks such as SMAC (https://www.automl.org/automated-algorithm-design/algorithm-configuration/smac/) were used on problems with more than 100 variables.”**
>
> To our knowledge, there is currently no BO algorithm that is better than DynaPPO or CbAS. DynaPPO (ICLR 2020) is benchmarked against BO approaches and outperforms it on some of the same landscapes we study. Additionally, we do perform a standard BO on the smaller TFbind landscape (see Fig. A1), allowing the BO to enumerate the whole space (importantly, by giving it no sample restriction). We show that the Evo-BO approach is better at least in that setting (where BO is easy to scale, although much slower than other approaches). The referee’s comment about adapting SMAC, a hyperparameter-tuning package for search algorithms, is interesting, but requires separate research.
> Adapting BO algorithms that outperform DynaPPO or CbAS is not what we set out to do, but a valuable (and challenging) problem to solve. Additionally, sequence design is often done for sequences much larger than 14. We use 14 as one case study since we can enumerate the space fully. In our experience, standard BO simply doesn’t scale too well without domain-specific simplification.
>
>
> **“The authors should make more explicit why is this evolutionary algorithm a novelty, how does that improve wrt EAs?”**
>
> AdaLead is a simplified Wright-Fisher process, a fundamental model of evolution. This simplification allows for easier implementation (doesn’t require an explicit fitness function) and also a lower memory and performance burden (sampling only happens from a fraction of the population instead of the full population).  We have aimed to build a benchmark that can be easily reproduced, and used within FLEXS to develop better algorithms. We hope that by implementing FLEXS and providing simple benchmarks within (AdaLead being the simplest and most competitive), we discipline future studies to refrain from becoming increasingly complex without checking performance against simple approaches; here we put forward one such approach that (to our surprise) outperforms the state of the art.

---

> > ### Author Response · Authors · 2020-11-15
> > **Response to minor comments**
> >
> > Minor comments
> >
> > **Figure 2 is not cited in the text.**
> >
> > Great point, we will cite it, thank you.
> >
> > **CMA-ES is not introduced - even if the CMA-ES is a popular algorithm it should be properly introduced.**
> > That is a helpful suggestion. We wrote an explanation of CMA-ES in the SI but will add more about it in the main text.
> >
> > **Model-free evolution in Figure 2 is not introduced. What is that algorithm about?**
> > Great point. We will introduce this. It is a standard Wright-Fisher process without the assistance of model predictions.
> >
> > **Figure 2, I'm not sure what is the y-axis in the performance plot. In these types of papers, we usually show simple regret or similar performance quantities. It seems that the y-axis is the higher the better and that 1 is the max achievable?**
> > 	Thank you for pointing this out. The y-axis of our plots is the maximum fitness of all sequences proposed so far by the algorithm. We normalize fitnesses to be between zero and one, making 1 the max possible fitness, and seek to propose sequences with high fitness. We mention this in the paper, but will make the labeling more clear.
> >  We plot the max proposed fitness because there is no proper notion of regret in this setting. Regret is the difference between realized score and the best score that would have been achieved by picking, in hindsight, the best possible action. While there is an iterative structure to our problem setting, it is not stochastic, meaning that the best possible action would simply be to propose the sequence with the highest possible score. This is bounded by one, the max possible fitness, giving us our original plot.
> >
> > **Figure 2 should give a better understanding of what the figure means and what are the methods that perform best by writing about that in the text of the paper.**
> > Good point. We will expand more upon the algorithm performances shown in Figure 2 in Sections 3.1 and 3.2.
> >
> > **Not clear to me why the queries to the surrogate model are sample-restricted. This is an uncommon setting in Bayesian optimization where usually the budget is restricted for the evaluation of the latent black-box function.**
> > 	The sample restricted setting is relevant for three reasons:
> > - (i) In order to study these algorithms on small landscapes that can be fully enumerated (which is relevant to compute peaks and study algorithm quality), but remain faithful to the spirit of the problem (where the sequence space is so large that it cannot be enumerated computationally), we have restricted the number of virtual samples (except in a case study for BO as explained above Fig. A1). If we did not do that, for alpha=1, TFbind and RNAL14 landscapes can be solved in a single batch (by doing 10^5 and 10^7 virtual queries respectively).
> > - (ii) In practice, while virtual queries are much cheaper than ground-truth queries they are not free. Adding a sample restriction ensures that we can measure the proposed algorithms based on their efficiency.
> > - (iii) This helps us put all algorithms on equal footing in terms of queries made to the oracle. In principle this should hurt evolutionary approaches the most, as their generative process is stochastic and often wasteful (unlike VAE in CbAS/DbAS and the policy network in DynaPPO, which are trained models). Our experiments with increasing the virtual screening ratio (i.e. number of virtual queries per ground truth sample) to up to 10000 on specific landscapes are consistent with our current results (most algorithms converge before using the full budget). Due to the massive slowdown that occurs with these higher ratios however, we chose not to reproduce all of our few hundred experiments with high virtual screening ratios; in general, it would take a prohibitively long time to do so (with some experiments taking on the order of weeks to complete).
> >
> > **Not entirely clear what the role of the cardinality |S| in the optimization is. Is this an optimization with unknown feasibility constraints problem formulation?**
> >
> > We include this to simply point out that we do not know if there are any or many solutions with score above $y_\tau$. A good algorithm can find more solutions, hence is better at optimization in our context, because as we explained in the text, finding multiple solutions are helpful to hedge risk in downstream tasks. We are happy to clarify this further or change our presentation.

---

### Official Review · AnonReviewer4 · 2020-10-28
**An empirical study on sequence design&optimization that might not be suitable for ICLR**

**Rating:** 4
**Confidence:** 4

**Review:**

In this work. the authors addressed a critical issue on biological sequence design. It is an important task in biomedical research. The authors proposed an evaluation/simulation framework along with an evolutionary algorithm and showed the simple evolutionary algorithm "AdaLead" outperformed several existing methods. The paper is clearly written.

pros:
1. Addressing an important issue. Sequence design and optimization are critical for many real-world problems, such as antibody optimization, etc.
2. The authors have surveyed and tested many existing algorithms. They have also considered many settings such as DNA, RNA, and protein as well as "swampland" where the optimization landscape is flat. Experiments are done thoroughly.
3. Offered an open-sourced sandbox for model comparisons and proposed and evaluated a simple yet strong-performing evolutionary algorithm that should be used as a baseline in further works in this space.

cons:
1. First of all, although I appreciate a thorough investigation and comparison between existing methods and implementing an evolutionary algorithm as a strong baseline, it is, however, unclear to me the ML significance of this work to the ICLR community. It might be more suitable for a specific compbio venue.
2. The authors proposed an evaluation framework where a noisy version of the oracle learned from sampling, $\phi^{'}$, could be obtained. Although this is a valid and reasonable framework, it is not fully in line with other algorithms' assumptions. For example, DbAS and CbAS are more conservative and exploring locally by generating a new batch of sequences that are similar to the best performing one in the previous batch. This can be also a valid approach as the extrapolation might be very bad. Moreover, the authors tuned AdaLead on their proposed evaluation framework and claimed it is not sensitive to parameters, I do not see any results regarding this matter.  In short, I suspect the evaluation framework proposed here is biased towards a simple evolutionary algorithm that does not even use much information (as AdaLead outperforms other more complicated evolutionary algorithms in fig A1B).
3. Related to 2, there is a lack of exploration to understand why AdaLead is outperforming. What could we learn from this result?

minor:
1. The authors talked about batch size issues in the paper. Could they give more details or some results on this?
2. Some of the figure labels can be improved. There are some acronyms likely not familiar to computer scientists with little biological background left unexplained, such as "w.t." in fig 1B.

---

> ### Author Response · Authors · 2020-11-15
> **Reponse to major comments**
>
>
> We thank the reviewer for helpful remarks and criticisms. We would appreciate specific advice on how we can improve the paper to satisfy their criteria and are eager to improve the paper with their feedback.
>
> -------------------
> **“First of all, although I appreciate a thorough investigation and comparison between existing methods and implementing an evolutionary algorithm as a strong baseline, it is, however, unclear to me the ML significance of this work to the ICLR community. It might be more suitable for a specific compbio venue.”**
>
>
> As the reviewer noted, sequence design is an increasingly studied issue in ML. The two SOTA algorithms we compare against are published in ICML 2019 (as a talk, CbAS) and ICLR 2020 (DynaPPO). We are targeting this work to the same audience. The purpose of FLEXS is to unify a testbed where ML researchers can readily compare their new algorithms with what’s been done. The second contribution of the paper is AdaLead, which is simple by design. We try to make a point here that ML researchers should benchmark against simple algorithms before developing complex ones that are ultimately harder to use in practice, are harder to replicate, and will not outperform this baseline. We think it would be relevant to the ML community to be aware of this result as they develop increasingly complex methods which might never get tested against “boring” or “simple” benchmarks, even though those benchmarks are strong and more accessible in practice.
>
>
> **“The authors proposed an evaluation framework where a noisy version of the oracle learned from sampling, ϕ′, could be obtained. Although this is a valid and reasonable framework, it is not fully in line with other algorithms' assumptions. For example, DbAS and CbAS are more conservative and exploring locally by generating a new batch of sequences that are similar to the best performing one in the previous batch. This can be also a valid approach as the extrapolation might be very bad.”**
>
> The cases of $\phi’$ where $\alpha=0$ and $\alpha=0.5$, which we study, are in fact where extrapolation is very poor. Even then, AdaLead is comprehensively outperforming CbAS and DbAS (See Fig. 2B).  The results are similar for empirical models where the ground truth is not accessed directly, only through measurements, which is in line with other algorithm assumptions.
>
> **“Moreover, the authors tuned AdaLead on their proposed evaluation framework and claimed it is not sensitive to parameters, I do not see any results regarding this matter. In short, I suspect the evaluation framework proposed here is biased towards a simple evolutionary algorithm that does not even use much information...”**
>
>
> We picked our challenge landscapes based on whether or not we found them relevant and whether they can be studied in a thorough manner. It would be scientifically dishonest of us to pick our challenges so that a particular algorithm performed well. We were surprised by the relative performance of AdaLead as we added new challenges to FLEXS. We would like to point out that we use the TFbind landscape used by Angermueller et al. 2020 as one of the benchmarks, and reproduce the performance of DynaPPO (and show that AdaLead is competitive). Overall, FLEXS includes challenges that are biologically realistic, more numerous, and on par with other papers in terms of their challenge (if not more difficult).
>
> Regarding parameter sensitivity, it is shown for one landscape in Fig. A2; we are happy to provide further verification and invite the referee to suggest additional checks.
>
> **“Related to 2, there is a lack of exploration to understand why AdaLead is outperforming. What could we learn from this result?”**
>
> It is difficult to gauge why AdaLead outperforms the other algorithms, because we compare against algorithms which are quite complex. We suspect that biological sequences have strong correlation structure within neighborhoods, and AdaLead’s simple trick to start from sequences which have already performed well ensures robustness. The rest of the performance comes from a combination of simple hill-climbing with adaptivity in flat areas of the optimization surface. It is also surprising to us that this is enough to beat the other approaches.  AdaLead could of course be improved by modifying the generative process that makes the variants (e.g. using a VAE like CbAS), but we explicitly refrained from doing so to keep the algorithm as a simple, reproducible benchmark.

---

> > ### Author Response · Authors · 2020-11-15
> > **response to minor comments**
> >
> > Minor points
> > **“The authors talked about batch size issues in the paper...”**
> >
> > We will add results from smaller batch sizes in the appendix. They make no changes in our results. The reason we have not included larger batch sizes is that some of the algorithms in our study (not AdaLead) are hard to use with larger batch sizes (they converge without proposing enough diversity or take prohibitively long to do so).
> >
> > **“...There are some acronyms likely not familiar to computer scientists with little biological background left unexplained, such as "w.t." in fig 1B.”**
> >
> > Thank you for pointing this out. “w.t.” stands for “wild-type”, which represents the starting sequence. We will be more clear in the updated manuscript.

---

### Official Review · AnonReviewer3 · 2020-10-28
**Limited methodological contribution, unclear impact of experimental design choices**

**Rating:** 5
**Confidence:** 2

**Review:**

In this work, the authors propose a greedy search approach for designing biological sequences in an active learning, batch setting. The algorithm is a fairly standard evolutionary algorithm which identifies the set of candidates at each iteration by adapting the best candidates from the previous iteration, and then evaluating those adaptations using a surrogate model. A set of experiments (using an evaluation sandbox also proposed by the authors for this work) suggest the proposed approach finds more high-quality solutions than competing approaches; however, especially in the experiments using the realistic/empirical surrogate model (an ensemble of 3 CNNs), the quality of the best solutions found by several approaches are statistically similar.

As promised by the authors, the proposed approach indeed appears rather straightforward to implement. Further, Table 1 does show that the proposed approach identifies more sequences which meet the specified criteria. To the best of my knowledge, the paper also places the work into context with respect to existing work.

As a non-expert approaching this paper, I find it very difficult to gauge the impact of the choice of the abstract and null models in the experiments. In particular, all of the results (and especially Table 1 and Figure 2B) suggest that the proposed approach is much better than the competition when using these models. On the other hand, when the ensemble is used as the surrogate model (e.g., Figures 2C(bottom) and 2D(left)), the performance of several methods become very similar.

Since the methodological contribution of this work is rather limited (again, a pretty standard evolutionary algorithm), it is concerning to me that the strongest empirical results only arise in the settings based on these choices.

Concerning the diversity of solutions found, in Table 1, it is not clear to me how meaningful the diversity measure (number of local optima with a score above the given threshold) actually is. Especially in cases where many solutions are found, it would be useful to also characterize the diversity of the sequences themselves, for example, by using BLOSUM or some other sequence similarity measure.

Of course, actual wetlab validation demonstrating the superiority of the proposed approach would significantly strengthen the contribution.

Minor comments
----------------

The references are not consistently formatted.

Table 1 should include some measure of variance.

When introducing “optimization”, the authors rightly point out that, e.g., moderate binding may be preferable to stronger binding in some cases. However, they also suggest that the stronger binder has a better fitness. To me, this suggests the model for fitness is wrong and fits better with the “robustness” theme (since an accurate fitness model would have a parabola shape or some such in this example). It is also not clear to me whether any of the objective functions for the simulations capture this phenomenon.

I found the description of the RNA ground truth simulation unclear.

---

> ### Author Response · Authors · 2020-11-15
> **Response to major comments**
>
> We thank the reviewer for their multiple helpful comments, and would like to ask if there are specific improvements we can offer to alleviate their concerns about suitability of the paper. We also have some responses to your specific concerns below.
>
> We appreciate that the reviewer is concerned about the technical contribution within AdaLead;
> The primary contribution of our paper, in our view, is to build an environment that enables disciplined benchmarking on relevant biological challenges. As a component of the benchmarking environment, we show that a simple algorithm (AdaLead) is more robust and as performant (if not better) than SOTA. This was an unexpected ‘discovery’ when we implemented AdaLead as a simple baseline.  We aim to communicate that a simple, easily reproducible algorithm such as AdaLead can serve as a good benchmark before turning to far more complex approaches or developing new methods. We hope that the presence of a strong and robust baseline along with a benchmarking environment will encourage further research and ultimately lead to better machine-guided design algorithms.
>
> **“....however, especially in the experiments using the realistic/empirical surrogate model (an ensemble of 3 CNNs), the quality of the best solutions found by several approaches are statistically similar.”**
>
> We ask the reviewer to kindly clarify this remark because Figure 2A, C and Table 1 all show that the best solutions found by AdaLead with 3-CNN are clearly better than those found with other published SOTA algorithms (CbAS/DynaPPO).  The only algorithms that on occasion perform closely to AdaLead are also evolutionary algorithms that have not been used as benchmarks for this problem, yet AdaLead is simpler. Moreover, we have found it to be consistently good in many challenges (optimization, robustness, consistency, across TFbinding, RNA and protein landscapes), which is not the case for other algorithms (e.g. Evo_BO and CMAES are less consistent, less robust, and less performant (or at best occasionally match) AdaLead).
>
> **“Since the methodological contribution of this work is rather limited (again, a pretty standard evolutionary algorithm), it is concerning to me that the strongest empirical results only arise in the settings based on these choices.”**
>
> The choice of the abstract models is meant to separate the performance of the algorithm from that of the model: (i) alpha=1, when the model is perfect, (ii) alpha=0, when the model is completely noisy and therefore uninformative, and (iii) various alphas between (0,1) which allows us to study consistency (relationship between model quality and outcomes).  We observe that the performance of empirical models is quite similar to those captured by abstract models with alpha in the range of 0.5-0.95. However, as we mentioned to Reviewer 1 above, for all empirical models we have tried (which includes other classes such as Random forest, Linear regression, ...) we find the same order of performance between AdaLead and other published methods.
>
> **“Concerning the diversity of solutions found, in Table 1, it is not clear to me how meaningful the diversity measure (number of local optima with a score above the given threshold) actually is. Especially in cases where many solutions are found, it would be useful to also characterize the diversity of the sequences themselves, for example, by using BLOSUM or some other sequence similarity measure.”**
>
> The number of local peaks found is relevant as we worry that sequences are stuck at local peaks, and cannot explore to find new ones. It is more meaningful than finding many points that are distinct but all are rather poorly optimized. We are happy to report more diversity statistics for our top sequences in the final manuscript. We would note that BLOSUM would only work in the protein case.

---

> > ### Author Response · Authors · 2020-11-15
> > **Response to minor comments**
> >
> > Minor comments
> >
> > **“The references are not consistently formatted.”**
> >
> > We will fix the formatting, thank you for pointing this out.
> >
> > **“Table 1 should include some measure of variance.”**
> >
> > We will add variance to the table.
> >
> > **“When introducing “optimization”, the authors rightly point out that, e.g., moderate binding may be preferable to stronger binding in some cases. However, they also suggest that the stronger binder has a better fitness. To me, this suggests the model for fitness is wrong and fits better with the “robustness” theme (since an accurate fitness model would have a parabola shape or some such in this example). It is also not clear to me whether any of the objective functions for the simulations capture this phenomenon.”**
> >
> > Thank you, we will clarify in the paper. Our inclusion of the moderate binding example is more of a biological point for ML researchers, rather than one to worry about in our optimization. One can always transform the optimization objective to one that is maximized or minimized when the binding is at the desired value (and scores poorly if it binds too strongly). We simply wanted to say that maximization is not always the necessary “biological” objective, even though we simplify the problems to have this property in this paper. We could choose a different objective, and then define a transform to end up with the same answer.
> >
> > **I found the description of the RNA ground truth simulation unclear.**
> >
> > We will add more details to the simulator description. In most cases, the ground truth simulator simply computes the binding energy between a sequence and one or two hidden targets.

---

### Official Review · AnonReviewer1 · 2020-10-30
**ADALEAD+FLEXS for sequence design problems**

**Rating:** 6
**Confidence:** 2

**Review:**

The authors implement an open-source simulation environment FLEXS to emulate complex biological landscapes (TF binding landscapes) and to train and evaluate (RNA or protein) sequence design problems. ‘Swampland’ landscapes with large areas of no gradient, are also studied.
They also propose a simple greedy-search evolutionary algorithm, Adapt-with-the-leader (ADALEAD) that is shown to perform better than its competitors and to optimize in challenging settings.

The paper is well written and motivated and previous, relevant literature is discussed.

Comments:
>> An important component of the environment and for ADALEAD is phi^prime. The paper mentions that this is fully supervised. How is one to come up with the appropriate	phi_prime for a problem at hand.

>> For the experiments, are the Ensembles always a 3-CNN model? Also, are the CNN architectures always the same?

>> Can non-neural net architectures be used as ensembles?

>> In the Figure 1A, it seems that the schema is iterative between the 4 layers shown by the arrows. Is this the case? If so, the iterative aspect is not mentioned clearly in the paper.

>> What is EI in the Bayesian optimization subsection?

---

> ### Author Response · Authors · 2020-11-15
> **Clarifications to review points**
>
> We thank the reviewer for their comments. Below are our responses to specific points.
> We would also appreciate it if the referee can explain their reasoning for giving us a “marginal” score, as we would like our work to be a useful resource for benchmarking, and the current score suggests that the referee has reservations about our work. We are eager to address your concerns. Specific responses to concerns are below.
>
>  **An important component of the environment and for ADALEAD is $\phi'$. The paper mentions that this is fully supervised. How is one to come up with the appropriate phi_prime for a problem at hand.**
>
> AdaLead or other algorithms can work with any regression model $\phi’$. The focus for making this sandbox is not for coming up with excellent models ($\phi’$), but to provide algorithms that can work with any model; hence we test a variety of $\phi’$ (which we implement). There are two flavors of $\phi’$: first are noise corrupted versions of the ground truth, and second are empirical models which learn from data (would be used in a real-world setting). The goal of FLEXS is for the user to be able to test their algorithm with any quality of $\phi’$ they wish, and see how it affects performance. The purpose of the noise-corrupted phi’ is to let the user study the effect of model quality. Additionally,  we show that using empirical models (like the CNN) does not change the relative strength of AdaLead, and this pattern is consistent for all of the landscapes we have tried.
>
>  **For the experiments, are the Ensembles always a 3-CNN model? Also, are the CNN architectures always the same?**
>
> We use a 3-CNN ensemble model (consisting of 3 independently trained CNNs with the same architecture) as a representative in the paper as it was the best performing ensemble we tried. However, it is trivial to run FLEXS with an ensemble of any size with a variety of model classes -- all the user has to do is to pick a set of model classes from our model catalogue in the code and pass them as arguments (e.g. Random forest, Linear regression, Global epistatic models, GP regressors). It is also easy to add any general class of models which isn’t already implemented in FLEXS.
>
> **Can non-neural net architectures be used as ensembles?**
>
>  Yes, and a mixture of different model classes can be ensembled together.
>
> **In the Figure 1A, it seems that the schema is iterative between the 4 layers shown by the arrows. Is this the case? If so, the iterative aspect is not mentioned clearly in the paper.**
>
> Thank you for pointing this out. There are in fact iterations between measurement, modeling, and exploration. We mention this, but perhaps the connection to the figure was unclear. We will address and clarify this in the following manuscript.
>
> **What is EI in the Bayesian optimization subsection?**
>
> Great point. EI stands for “Expected improvement” in Bayesian Optimization, which is an acquisition function commonly used in this setting.

---

### Author Response · Authors · 2020-11-15
**General points in response to reviews**

We thank the reviewers for their feedback. In light of certain concerns of the reviews regarding “novelty” of an evolutionary algorithm, we would like to make a general clarification.
The primary contribution of our paper, in our view, is to build an environment that enables disciplined benchmarking on relevant biological challenges. As a component of the benchmarking environment, we show that a simple algorithm (AdaLead) is more robust and as performant (if not better) than SOTA. This was an unexpected discovery when we implemented AdaLead as a simple baseline. In line with our objective for disciplined benchmarking, we aim to communicate that a simple, easily reproducible algorithm such as AdaLead can serve as a strong and robust baseline for developing more complex algorithms. In our view such an environment is essential in enabling disciplined ML research, as sequence design is a very active area of research at the intersection of biology and ML.
As the methods we compare against in this work are published in ML conferences, we think that our work is most suitable for the same audience, which is interested in developing model-guided sequence design algorithms. We introduce specific criteria in this context (such as robustness and consistency) that are directly relevant to model-guided optimization, and provide tools for their study.
Aside from our clarifications, we have the following plan to improve the paper based on reviewer suggestions:
- We will reorganize the paper to better communicate the message encoded above.
- We will provide more detailed explanations of why we think AdaLead is (1) a competitive benchmark, (2) relevant in this context.
- We will add additional supporting information/experimental results that should alleviate referee concerns.

We are eager to incorporate any suggestions to this work in order to achieve the aim described above, and in fact invite further actionable feedback from reviewers to make this work serve the described purpose.

---

### Decision · Program_Chairs · 2021-01-07
**Final Decision**

**Decision:**

Reject

**Comment:**

This paper considers the problem of (biological) sequence design and optimization. The authors made an interesting yet important case that in certain sequence design tasks, a simple evolutionary greedy algorithm could be competitive with the increasingly complex contemporary black-box optimization models.

Most reviewers appreciate the design of the open-source simulation environment in benchmarking AdaLead (and other competing algorithms) in a number of biological sequence design tasks (e.g. TF binding, RNA, and protein). However, there is a common concern that the experimental results are not fine-grained enough to explain the outperforming results of the proposed algorithm. There are also unresolved comments on missing important BO baselines in the empirical study. As a purely empirical work, these appear to be important concerns. While these results appear to be useful for the domain of biological sequence design, the reviewers are unconvinced that the proposed algorithm is significantly novel, or the results are sufficiently compelling to merit an acceptance to this venue.